# Age and Emotional Distress during COVID-19: Findings from Two Waves of the Norwegian Citizen Panel

**DOI:** 10.3390/ijerph18189568

**Published:** 2021-09-10

**Authors:** Line I. Berge, Marie H. Gedde, Bettina S. Husebo, Ane Erdal, Camilla Kjellstadli, Ipsit V. Vahia

**Affiliations:** 1Centre for Elderly and Nursing Home Medicine, Department of Global Public Health and Primary Care, Faculty of Medicine, University of Bergen, 5009 Bergen, Norway; marie.gedde@uib.no (M.H.G.); bettina.husebo@uib.no (B.S.H.); ane.erdal@uib.no (A.E.); camilla.kjellstadli@fhi.no (C.K.); 2Olaviken Gerontopsychiatric Hospital, 5306 Erdal, Norway; 3Haraldsplass Deaconess Hospital, 5009 Bergen, Norway; 4Department of Nursing Home Medicine, Municipality of Bergen, 5020 Bergen, Norway; 5Department of Cancer Treatment and Medical Physics, Haukeland University Hospital, 5021 Bergen, Norway; 6Division of Health Registry Research and Development, Norwegian Institute of Public Health, 5808 Bergen, Norway; 7Division of Geriatric Psychiatry, McLean Hospital, Belmont, MA 02478, USA; ivahia@mclean.harvard.edu; 8Harvard Medical School, Boston, MA 02115, USA

**Keywords:** emotional distress, COVID-19, outbreak, older adults, age, gender, health, income

## Abstract

Older adults face the highest risk of COVID-19 morbidity and mortality. We investigated a one-year change in emotions and factors associated with emotional distress immediately after the onset of the pandemic, with emphasis on older age. Methods: The online Norwegian Citizen Panel includes participants drawn randomly from the Norwegian Population Registry. Emotional distress was defined as the sum score of negative (anxious, worried, sad or low, irritated, and lonely) minus positive emotions (engaged, calm and relaxed, happy). Results: Respondents to both surveys (*n* = 967) reported a one-year increase in emotional distress, mainly driven by elevated anxiety and worrying, but we found no difference in change by age. Multilevel mixed-effects linear regression comparing older age, economy-, and health-related factors showed that persons in their 60s (ß −1.87 (95%CI: −3.71, −0.04)) and 70s/80s (ß: −2.58 (−5.00, −0–17)) had decreased risk of emotional distress relative to persons under 60 years. Female gender (2.81 (1.34, 4.28)), expecting much lower income (5.09 (2.00, 8.17)), uncertainty whether infected with SARS-Cov2 (2.92 (1.21, 4.63)), and high self-rated risk of infection (1.77 (1.01, 2.53)) were associated with high levels of emotional distress. Conclusions: Knowledge of national determinants of distress is crucial to tailor accurate public health interventions in future outbreaks.

## 1. Introduction

Pandemic events represent global public health disasters, and their negative impact is exacerbated by detrimental effects on mental health [1,2]. At these times, it is important to identify people who face the highest risk of psychiatric disorders [3,4] and challenges around successful management of stress in order to tailor accurate public and mental health interventions [5,6,7]. Emotional distress refers to the negative emotional state characterised by physical and/or emotional discomfort [8] and existing literature is equivocal on how older adults’ mental health is influenced by pandemics. On the one hand, they might be at higher risk of deteriorating mental health because of their high risk of disease morbidity and mortality [9]. Long periods of physical isolation could further disproportionally increase the risk of social isolation and loneliness for persons less familiar with digital communication, such as home-dwelling elderly with dementia [10,11]. In the National Social Life, Health, and Ageing Project, Santini et al. followed more than 3000 people aged 57–85 years over a decade and showed that social disconnectedness predicted perceived isolation, which, in turn, led to higher symptoms of anxiety and depression [12]. During the 2003 SARS outbreak in Hong Kong, the one-year suicide rate in the elderly increased by 32%, yielding a rate of 37.5/100,000 among persons ≥ 65 years [13]. The increase was highest among females, and qualitative studies suggested that older suicide victims feared disconnection and were concerned that they contracted the disease [14]. However, evidence also exists to suggest the opposite phenomenon, where older adults are better placed to withstand the strains of disaster events. In the aftermath of natural disasters in Australia, older adults showed high resilience and social capital, which was regarded as a benefit for the whole society [15]. An important aspect for wellbeing among older adults in times of crisis is the trait of decreased emotional reactivity relative to younger adults, as demonstrated by Schweizer et al., utilising data from the Cambridge Centre for Ageing and Neuroscience cohort [16]. Results from the initial and intermediate phase of the COVID-19 outbreak suggest that older adults in high-income countries are less prone to stress symptomatology [1,17]; however, the extant data do not shed light on the interplay of risk and protective factors. While age has emerged as an important factor in determining resilience during the pandemic, we know little about its importance compared to other determinants of emotional distress, such as a perceived threat to the economy and health. Information on these relationships has implications for psychological interventions when facing community-wide stressors, and, above all, to ensure appropriate resource allocation. Based on the literature, we hypothesised that persons aged 60 years and older would experience less increase in emotional distress at the start of the pandemic and lower levels of distress relative to younger ones in the first weeks of the outbreak, also when accounting for other determinants for distress. In this study, we investigated a one-year change in emotions in a representative sample of the adult Norwegian population between spring 2019 and 20–29 March 2020, immediately after the COVID-19 lockdown on 12 March, comparing persons in their 60s, 70s, and 80s with those under 60 years. Further, we explore age in relation to other demographic-, economic-, and health-related factors associated with emotional distress immediately after lockdown.

## 2. Materials and Methods

The Norwegian Citizen Panel (NCP) (Table 1) was established in 2013 as an online research panel of approximately 25,000 inhabitants in Norway [18], constituting a representative sample of the adult Norwegian population of 4.2 million [19]. It aims to provide longitudinal data to study attitudes and opinions in the general population regarding Norwegian society and politics, including issues on trust, welfare, environment, and public health. NCP is operated by the Digital Social Science Core Facility (DIGSSCORE), an infrastructure for advanced social science data collection and multi-disciplinary research at the University of Bergen [20]. The collection and management of data are handled by Ideas2Evidene—a commercial entity in the private sector [21].

To yield high representativeness, random samples of residents aged 18 years and older have been drawn in 6 waves from the Norwegian Population Registry [22] after a personal invitation from 2013 onwards. A lottery for a travel gift card with a value of 25,000 NOK serves as an incentive for participation in each round. Surveys are conducted two times a year, and the 15th wave was undertaken from 21 March to 10 June 2019. Eight days after the Norwegian lockdown on 12 March 2020, the NCP Fast Track COVID-19 wave was distributed. Non-responders received reminders on the 25 and 27 March, and the survey closed on the 29 March 2020.

Measures: We measured self-reported levels of eight emotions during the preceding seven days, five were negative (anxious, worried, sad or low, irritated, and lonely) and three positive (engaged, calm and relaxed, happy). Each item score ranges from 0 to10, with higher scores indicating more intense emotion. We defined the main outcome ‘emotional distress’ as the difference between the sum score of the negative emotions minus the sum score of the positive emotions, yielding a total score with a range of −30 to 50. A high score indicates a high level of distress. Changes in emotions and emotional distress were defined as score differences between spring 2019 and the COVID-19 wave in March 2020.

We categorised age into four groups: below 60 years (reference group), 60–69 years, 70–79 years, and 80 years and older in 2019. Demographic covariates assessed in earlier waves of NCP included gender and educational level (primary school, high school, and college/university). We assessed data in the COVID-19 wave March 2020 on change in work situation (yes/no) and expected household income in 2020 (much lower, lower, higher, much higher and no change (reference group)), in addition to data on how respondents rated the importance of the information provided in press conferences from the government (1–5, a high score indicates high importance). Finally, we utilised health-related data assessing if the respondent was uncertain whether infected with SARS-Cov2 (yes/no), considered oneself or cohabitant vulnerable for infection (yes/no), self-rated health (1–5; a high score indicates good health), self-rated risk of infection (1–5: a high score indicates high risk), contentment with life (1–5; a high score indicates a high level of contentment), and confidence in others (1–10; a high score indicates a high level of confidence).

Sample and statistics: This study includes participants with complete responses on the outcome measure emotional distress both in spring 2019 and the COVID-19 wave March 2020 (Figure 1). We estimated the percentage of participants with a negative change in emotions, defined as an increase in the level of negative and/or a decrease in the level of positive emotions, and compared differences between age groups with logistic regression. Similarly, we estimated mean change in the level of emotions comparing differences between age groups with linear regression. Factors associated with emotional distress in the COVID-19 wave in March 2020 were explored with multilevel mixed-effects linear regression with region (county) as a random effect. The Akaike information criterion guided model selection, including the selection of variables. Finally, we applied multilevel mixed-effects linear regression with region as a random effect to evaluate associations between demographic factors and change in emotional distress. For both samples, the model showed a good fit when evaluated for robustness, multicollinearity, and heteroscedasticity.

We performed complete case analyses on the outcome measures emotions, and missing data on explanatory variables were handled by listwise deletion (missing range from 0–0.7%, except 15% on the level of education). The level of significance was 0.05 for all analyses. Descriptive statistics and regression modelling were performed with Stata version 16 [23], and figures were made in SPSS version 25 [24].

Ethics, regulations, and data protection: All respondents electronically signed informed consent before participation in each wave. Norwegian Centre for Research Data (NSD) has authorised the collection and storage of data in NCP (Project Number 118868). The Regional Committee for Medical and Health Research Ethics has approved the utilisation of data for health-related purposes (REK Vest Project Number 136825). A data protection impact assessment (DPIA) was developed to meet the requirements from European Union-wide law on data protection, (GDPR) (Ref 118868). Data are available from the authors upon reasonable request and with permission of The Norwegian Citizen Panel.

## 3. Results

### 3.1. Change in Positive and Negative Emotions and Emotional Distress from Spring 2019 to COVID-19 Wave in March 2020

In spring 2019, a random sample of 1347 participants were invited to assess levels of 8 types of emotion in the past 7 days, of which 1223 had complete responses. In March 2020, 12,051 participants were invited, and 11,443 had complete responses on the items assessing emotions. In total, 967 respondents had complete responses on both waves; these constitute our study sample (Figure 1).

Overall, 59.3% of the respondents experienced increased emotional distress from spring 2019 to March 2020 (data not shown), yet there was no difference in percentage change between the age groups (Table 2). The highest share of participants reported increased anxiety, and the lowest share reported increased loneliness. Relative to persons younger than 60 years, less increase in irritation was found among persons in their 60s (OR 0.66 (95% CI 0.48, 091) and 70s (OR: 0.45 (95% CI: 0.30, 0.66) *p* < 0.001), while a lower percentage of participants older than 80 years reported increase in feeling sad or low (OR: 0.39 (95% CI: 0.17, 0.93) *p* = 0.035).

Except for persons older than 80 years, all age groups experienced a significant increase in the mean level of emotional distress from spring 2019 to March 2020 (Figure 2a). This was driven by increased anxiety in all age groups, accompanied by increased worrying and reduced happiness among persons in their 60s, 70s, and 80s (Figure 2b). Additionally, less irritation and engagement were found among persons in their 70s, while persons in their 60s experienced increased loneliness. Overall, we found no difference between the age groups in mean change of the single emotions, the sum of positive and/or negative emotions, and emotional distress when evaluated with linear regression (data not shown). The only exception was less change in irritation and engagement among persons in their 70s relative to persons under 60 years (irritation: ß: −0.70 (95% CI: −1.16, −0.25), *p* = 0.002, engagement ß: −0.53 (95% CI: −0.98, −0.08), *p* = 0.020).

### 3.2. Relative Importance of Older Age Compared with Other Demographic-, Economic-, and Health-Related Factors on Level of Emotional Distress in the COVID-19 Wave in March 2020

Table A1 in Appendix A shows the distribution of emotional distress and covariates by age groups. Relative to persons under 60 years, older participants had lower levels of emotional distress, higher levels of education, and expected less impact on income and working situation. Table 3 shows demographics-, economic-, and health-related factors associated with emotional distress in the COVID-19 wave in March 2020 evaluated by multilevel mixed-effects linear regression with region as a random effect. Relative to persons younger than 60 years, the adjusted model showed that persons in their 60s experienced significantly less emotional distress (ß: −1.01 (95% CI: −1.58, −0.45), *p* < 0.001), while there were no significant differences between participants in their 70s and 80s. The opposite pattern emerged when conducting a sensitivity analysis categorising persons in their 70s and 80s as one combined age group, which also experienced less emotional distress relative to persons younger than 60 years (ß: −2.58 (95% CI: −5.00, −0.17), *p* = 0.036). Being female (2.81 (95% CI:1.43, 4.28), *p* < 0.001), expecting much lower household income in 2020 (5.09 (95% CI: 2.00, 8.17), *p*= 0.001), being uncertain whether infected with SARS-Cov2 (2.92 (95% CI: 1.21, 4.63), *p* = 0.001) and high self-rated risk of infection (1.77 (95% CI: 1.01, 2.53), *p* < 0.001) increased risk of emotional distress, while being content with life was strongly protective of emotional distress (−7.72 (95% CI: −8.78, −6.66), *p* < 0.001).

### 3.3. Relative Importance of Older Age Compared with Gender and Education Predicting Change in Emotional Distress from Spring 2019 to COVID-19 Wave in March 2020

We applied multilevel mixed-effects linear regression with region as a random effect to explore demographic factors assessed in spring 2019 associated with an increase in emotional distress. Female gender (ß: 2.61 (95% CI: 0.91, 4.31), *p* = 0.003) predicted an increase in emotional distress, while we found no effect of age groups and level of education (data not shown).

## 4. Discussion

Our primary aim via this study was to evaluate, in greater detail, the patterns and correlations of the emotional response to the acute stress of the COVID-19 pandemic in Norway using data from two waves of a population-based online research panel. This study augments the broader literature indicating that older adults have withstood the strains of the pandemic better than younger groups [17]. By exploring smaller age groups and subdomains of both negative and positive emotions, we confirmed our hypothesis of lower levels of emotional distress in the elderly relative to younger ones in the initial phase of the outbreak. We found that level of emotional distress was highest among females, persons expecting reduced income, those who reported high self-rated risk of SARS-Cov2, and additionally, those who were uncertain whether they were infected with SARS-Cov2. Driven by higher levels of anxiety and worrying, emotional distress increased in all age groups comparing March 2020 with spring 2019, and in contrast to our hypothesis, there were no differences in change by age. This suggests that older adults’ emotional response to the outbreak is similar to younger adults. We argue against down-prioritising psychosocial support to the elderly in times of outbreak but rather tailor interventions based on an evaluation of earlier levels of distress and economic- and health-related risks.

A range of studies have explored representations of emotional distress and its associated factors in diverging populations, applying a range of methods, definitions of emotional distress, and predictors. Searching PubMed in August 2021 revealed more than 20,000 results when employing search words ‘factors emotional distress’. In Table 4, we present a non-systematic collection of this literature to shed light on how our a priori selection of factors, viewed from a clinical perspective, aligns with previous work [25,26,27,28,29,30,31]. Several of the factors we found were associated with the level of distress immediately after the lockdown in the general Norwegian population, such as female gender, younger age, and expected lower income; these factors also evident as risk factors of deteriorating mental health in a recent systematic review of the impact of COVID-19 on mental health, with data from close to 100,000 participants from several continents [32]. This review additionally identified the presence of chronic/psychiatric illnesses, student status, and frequent exposure to social media/news concerning COVID as risk factors for distress. We also recognised two systematic reviews on persons with cancer [33] and multiple sclerosis [34] respectively, each concluding that baseline distress is a robust predictor of the development of emotional distress, underlining our conclusion that public and mental health interventions should account for earlier levels of distress.

Anxiety and worrying were the most accentuated emotions in our study, in line with the anticipated rise in COVID-19 health anxiety [35]. In the model quantifying predictors of emotional distress, female gender was strongly associated with distress, as well as the only demographic factor predictive of an increase in distress, compared to levels in spring 2019. This finding is consistent with a body of literature on gender discrepancies in anxiety and stress-related disorders [36], also evident during COVID-19 [37]. Relative to men, women have nearly twice the prevalence of generalised anxiety and panic disorder, as well as higher incidence and more severe symptoms of post-traumatic stress disorder after exposure to threatening events [38].

Compounding this, studies on mental health consequences of the current COVID-19 pandemic have shown the highest impact among persons with pre-existing vulnerability [39,40]. Investigating the prospective course of psychiatric disorders in three Dutch case–control cohorts, Pan et al. found a dose–response relation between the number and chronicity of depressive, anxiety, and obsessive-compulsive disorders and impact on mental health, fear of the virus, and poorer coping ability [39]. Furthermore, pre-existing mental health conditions, in addition to the female gender, younger age, and low education were predictive of higher levels of anxiety and depression in the UCL COVID-19 social study, prospectively following more than 70,000 people in the UK through the first months of the outbreak [40]. In this latter study, low income also predicted trajectories of the affective symptoms, in line with both our finding of an association between reduced expected income and emotional distress and the emerging literature on negative mental health effects of the economic uncertainty during COVID-19 [41,42]. As an illustration of less psychological vulnerability, we were not surprised to find that confidence in others and being content with life were protective against high levels of emotional distress, as these traits are considered core features of resilience [43].

Even though the highest levels of COVID-19-related emotional distress are evident among younger adults [32], elderly people are also affected, reflected by our findings of an increase in anxiety and worrying regardless of age during the first weeks of the outbreak. After three months of social distancing due to COVID-19 in the US, about 25% of 500 older adults responding to an online survey reported psychological distress, and participants with poor psychical health, low socioeconomic status, and low resiliency were at greatest risk [44]. On the other hand, a study following 776 adults on daily stressors for one week during the initial outbreak revealed that despite similar levels of perceived stress, older adults between 60–91 years reported less negative and more positive effect than younger ones [45], whereas participants older than 60 years reported the lowest level over time of anxiety and depressive symptoms in the UCL COVID-19 social study [40]. Yet, our findings of less emotional distress among older adults add vital nuances to the broader picture of higher resilience and lower stress reactivity among the oldest in the wake of the outbreak. Rather than exploring the impact of age by operationalising it as a continuous variable in multivariate analyses [39] or considering ‘old’ as persons above a certain age [40], we replicate the findings stratifying on 10-year groups while additionally exploring the relative importance of older age compared to other determinants of stress. This stratification also includes a distinct group of persons aged 80 years and older, participants less frequently included in population-based online surveys. In contrast to studies solely relying on psychometric compound measures, either assessing symptoms [40] or disorders [39], we here elaborate on the change in the individual positive and negative emotions over the course of one week. Interestingly, we found a one-year increase in loneliness only for respondents in their 60s, which contrasts with other findings from the UCL COVID-19 social study, in which age under 45 predicted the most severe trajectories of loneliness [46].

The main strength of this study is the utilisation of an established research framework. This enabled us to assess data from a large sample immediately upon the onset of the pandemic lockdown and compare results with a previous wave. The respondents were invited to the panel over years after several rounds of random selection from the Norwegian Population Registry, yielding high overall representativity. Nonetheless, our findings may not necessarily be generalisable to other countries and the oldest participants in the panel could be less representative of the older Norwegian population, as participation in an online panel requires high degree of cognitive resources and technological literacy. We are, therefore, at risk of including the healthiest elderly, possibly underestimating the level of emotional distress, particularly in the oldest age group. A further limitation is the scare documentation on construct validity and operationalising of the item assessing self-reported levels of the eight emotions during the past seven days. Ideally, we should have conducted a systematic search in the literature identifying factors associated with emotional distress in the general population before the outbreak. Instead, our regression models were built after the inclusion of what we, as clinical researchers, a priori considered as relevant explanatory variables assessed in the COVID-19 wave aiming at the best possible model fit. Yet, we are evidently not able to evaluate the impact of other highly relevant but unmeasured variables, such as living situation and perceived support. This is particularly relevant when exploring variables predicting an increase in distress between the waves, as only basic demographic data were available for the complete prospective sample.

## 5. Conclusions

Older persons experienced lower levels of emotional distress in the initial phase of the Norwegian COVID-19 outbreak relative to younger adults. Yet, the increase in distress was similar across age groups, nuancing previous reports on decreased emotional reactivity among older adults. We, therefore, argue against down-prioritising psychosocial support to the elderly in times of outbreak but suggest tailoring public and mental health interventions according to earlier levels of distress, also considering economic- and health-related risks. Future research should explore the long-term mental health impact of the outbreak and strategies for recovery in various populations [47,48,49]. In particular, groups of elderly people who do not readily respond to online surveys should be followed, including those with cognitive impairment and dementia.

## Figures and Tables

**Figure 1 ijerph-18-09568-f001:**
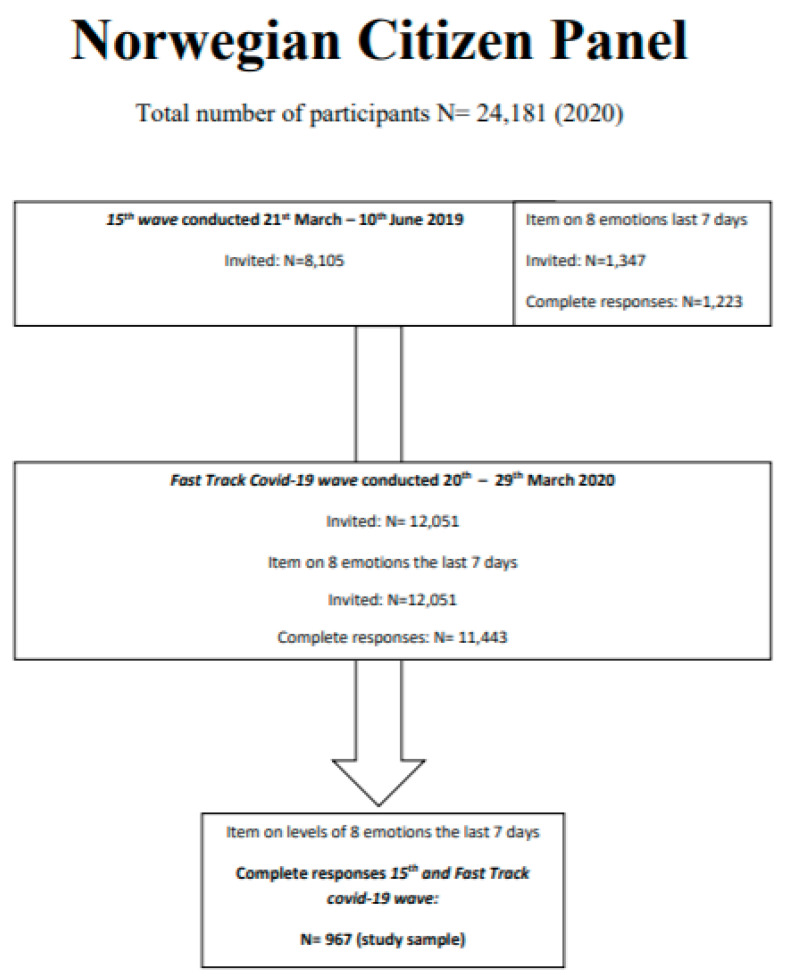
Flowchart of participants in the Norwegian Citizen Panel.

**Figure 2 ijerph-18-09568-f002:**
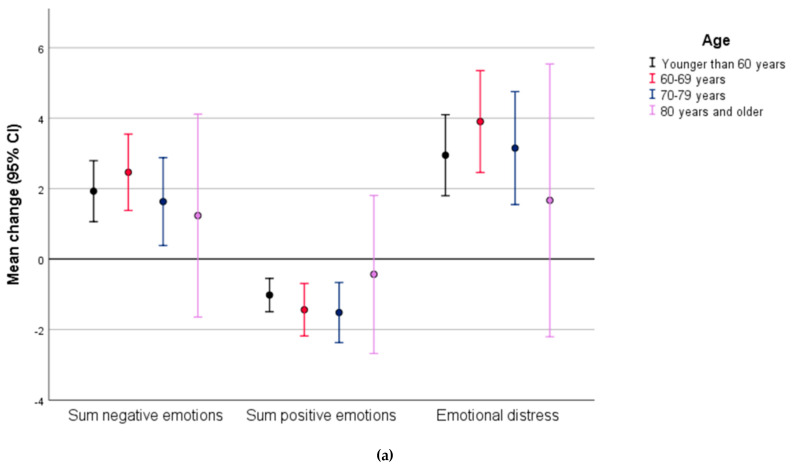
(**a**) Mean change in sum of emotions by age from spring 2019 to COVID-19 wave March 2020 (N = 967), range from −30 to 50. The zero line denotes ‘no change’ in emotions; (**b**) mean change in single emotions by age from spring 2019 to COVID-19 wave March 2020 (N = 967), range from −10 to 10. The zero line denotes ‘no change’ in emotions.

**Table 1 ijerph-18-09568-t001:** The Norwegian Citizen Panel.

Online social science research panel established in 2013
Approximately 25,000 respondents ≥ 18 years
Respondents invited after random selection from the Norwegian Population Registry
Surveys conducted two times a year on selected samples
Incentive for participation: lottery for travel gift card value 25,000 NOK
Operated by the University of Bergen, Norway

**Table 2 ijerph-18-09568-t002:** Percentage of respondents with negative change in emotions from spring 2019 to COVID-19 wave March 2020 in The Norwegian Citizen Panel (N: 967).

Last Seven Days, to Which Extent Did You Feel	<60 Years N = 514 % (Ref)	60–69 Years N = 255 %/OR/*p* *	70–79 Years N = 168 %/OR/*p* *	≥80 Years N = 30 %/OR/*p* *
Negative emotions:				
Anxious	56.4	56.1/0.99/0.928	52.4/0.85/0.361	43.3/0.59/0.165
Worried	50.3	46.3/0.85/0.283	48.2/0.92/0.625	46.7/0.86/0.692
Sad or low	43.3	40.0/0.86/0.344	35.1/0.70/0.054	**23.3/0.39/0.035**
Irritated	42.6	**32.9/0.66/0.010**	**25.0/0.45/<0.001**	36.7/0.78/0.523
Lonely	34.4	35.3/1.04/0.814	32.7/0.93/0.687	36.7/1.10/0.803
Positive emotions:				
Engaged	40.5	44.6/1.05/0.770	47.6/1.34/0.104	50.0/1.47/0.305
Calm and relaxed	41.1	39.6/0.94/0.701	39.3/0.93/0.686	40.0/0.96/0.909
Happy	55.5	53.7/0.93/0.651	51.2/0.84/0.336	43.3/0.61/0.199
Sum scores				
Sum of five negative emotions	56.8	53.7/0.88/0.418	53.0/0.86/0.385	53.3/0.87/0.709
Sum of three positive emotions	55.3	53.7/0.94/0.689	60.7/1.25/0.215	53.3/0.93/0.837
Emotional distress ^@^	58.2	60.4/1.10/0.556	61.9/1.16/0.393	53.3/0.82/0.602

% of participants with negative change in emotions; *—OR (odds ratio) and *p* value for comparison of percentage of participants with negative change in emotions from spring 2019 to COVID-19 wave March 2020, comparing persons 60–69 years, 70–79 years, and 80 years and older with persons 60 years and younger using univariate logistic regression.; ^@^—difference negative minus positive emotions. Bold format indicates significant differences between the age groups in negative change in emotions.

**Table 3 ijerph-18-09568-t003:** Factors associated with level of emotional distress in the COVID-19 wave in March 2020 of the Norwegian Citizen Panel (N: 967).

Fixed Effects	Beta (95% CI)	*p*
Age groups (ref born 1960 and later)		
1950–1959	−1.87 (−3.71, −0.04)	0.046
1940–1949	−2.34 (−4.82, 0.14)	0.064
1939 and earlier	−4.19 (−8.66, 0.27)	0.066
Gender (ref male)		
female	2.81 (1.34, 4.28)	<0.001
Level of education (ref primary school)		
High school	2.40 (−1.02, 5.82)	0.168
College/university	3.04 (−0.30, 6.38)	0.074
Expected household income in 2020 (ref no change)		
Much lower	5.09 (2.00, 8.17)	0.001
Lower	1.16 (−0.77, 3.09)	0.239
Higher	−1.23 (−5.34, 2.87)	0.556
Much higher	−7.60 (−21.69, 6.47)	0.290
Change in work situation (ref no)		
Yes	−0.18 (−1.90, 1.53)	0.834
Importance of press conference from government *	0.10 (−0.71, 0.92)	0.803
Uncertain whether infected by SARS-Cov2 (ref no)		
Yes	2.92 (1.21, 4.63)	0.001
Consider oneself vulnerable for infection with SARS-Cov2 (ref no)		
Yes	−1.31 (−3.32, 0.69)	0.199
Consider cohabitant vulnerable for infection with SARS-Cov2 (ref no)		
Yes	−1.64 (−3.37, 0.08)	0.062
Self-rated health *	1.32 (0.40, 2.34)	0.005
Self-rated risk of infection with SARS-Cov2 *	1.77 (1.01, 2.53)	<0.001
Content with life *	−7.72 (−8.78, −6.66)	<0.001
Confidence in others **	−0.31 (−0.65, 0.02)	0.066
Random effect	Var_cons	
County	1.72 (0.34, 8.69)	

Multilevel mixed-effects linear regression with region as a random effect and emotional distress as outcome, high level indicates high distress. Beta (95% CI) is the effect size of the variable on the outcome, the corresponding *p*-value in bold indicates significant change in effect size relative to the reference category. * range 1 (low)–5 (high), ** range 1 (low)–10 (high).

**Table 4 ijerph-18-09568-t004:** Pre-pandemic and pandemic studies on emotional distress and associated factors.

Author/Journal/Year	Design	Population (*n*)	Age	Outcome	Predictors	Comments
Viertio/BMC Public Health/2021/([25])	Cross-sectional	Finnish Regional Health and Well-being Study (*n* = 34,468)	20–65 years	Mental Health Inventory-5 (MHI-5)	Female gender, loneliness, job dissatisfaction, and family–work conflict	Protective factors: able to balance work and family life
Persson/Scan J Rheumatol 2005 [26]	Prospective	Early rheumatoid patients in Sweden (*n* = 158)	≥18 years	Symptom Checklist Scale (SCL-90)	Level of distress at baseline, female gender, young age, cohabiting, less social support	Disease activity weakly associated with distress
Løvstad/Disabil Rehabil 2020 [27]	Prospective	Survivors of terror attacks in Norway (*n* = 30)	19–71 years	Hopkins Symptom Checklist-25 (HSCL-8)	Neuroticism	Protective factors: resilience, optimism, social support. Injury severity not associated with emotional distress
Johnson/Injury/2019 [28]	Prospective	Patient admitted to a major trauma centre in the UK (*n* = 114)	All ages	CORE-10	High score on posttraumatic adjustment screen (PAS) at baseline, living outside hospital area.	No association between risk of distress development and sociodemographic factors and overall injury severity
Salvarani/Nursing Education Practice/2020 [29]	Cross-sectional	Nursing students affiliated with teaching hospitals in Italy (*n* = 622)	Young adults	GHQ-12, Italian version	Emotional regulation difficulties and empathic personal distress	No gender differences, senor students and students with high mindfulness score had lower distress.
Kabasawa/Plos One 2021 [30]	Cross-sectional COVID-19 sample	Workers in Japan (*n* = 609)	Adults	Kessler Psychological Distress Scale (K6)	Female gender, younger age, increased workload.	‘Staying at home’ regarded biggest life change.
Achdut/Int J Environ Res Public Health 2020 [31]	Cross-sectional COVID-19 sample	Young Israeli people (*n* = 389)	20–35	Modified items from the Israeli Social Survey (ISS)	Unemployment, financial strain, loneliness.	Protective factors were trust, optimism, and sense of mastery.

## Data Availability

Data are available from the authors upon reasonable request and with permission from the Norwegian Citizen Panel https://www.uib.no/en/citizen (accessed on 4 September 2021).

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
