# Peer review of "Age and Emotional Distress during COVID-19: Findings from Two Waves of the Norwegian Citizen Panel"

_ijerph, 2021, doi:10.3390/ijerph18189568_

Round 1
Reviewer 1 Report
The authors discussed an interesting topics which introduced the emotional changes during the COVID, especially for the elder people. In general, this study is well designed, but the presenting ways can be further improved. Therefore I will give the major revision for this paper:
1, In table 2, I guess the "OR" is odds ratio, which is not defined. Is the % in 2019 divided by % in 2020. or reversed ?
2, Also in that table, why the authors didn't show the OR and p values for <60 group.
3, I believe the figure 2 a & b can be moved to appendix, instead I suggested the author to find another data visualization method, and then show the most important data extracted from Figure 2.
4, In section 3.2, I am still confused how the authors set up the regression analysis, like in the method section, the author mentioned
"We estimated percentage of participants with negative change in emotions, defined as an increase in level of negative and/or a decrease in level of positive emotions, and compared differences between age groups with logistic regression. "
Do the author meaning by y=alpha +peta*x, where y is negative change, x is the positive change?
I stopped my review by the results part, will review the discussion part after the author clearly presented their results
Author Response
Please see the attatchment

Reviewer 2 Report
This paper deals with a very interesting topic, and the following revisions are required.
First, an extensive review of previous studies on emotional distress is needed. It is necessary to systematically review the independent variables that may affect emotional distress, and to examine whether these independent variables are reflected in Table 3.
Second, many studies on distress after COVID-19 have been conducted in relation to present study. It is necessary to explain the results by comparing the current research results with the previous research results.
Third, the age of the elderly is the main focus of this study. It is necessary to establish a hypothesis and statistically verify it after discussing at the theoretical model regarding distress in the elderly.
Fourth, in the conclusion part, it is necessary to suggest what points this study will theoretically contribute and whether future research is necessary.
Author Response
Please see attatchment

Round 2
Reviewer 1 Report
Appreciate for the revisions from the authors, now I do think the current version is acceptable to publish in IJERPH.
Author Response
Thank you
Reviewer 2 Report
There were so little changes in revised paper. Authors did not fully accept the review comment and therefore there still remains a lot of things revised for the sake of developing the paper.
Author Response
Reviewer 2:
There were so little changes in revised paper. Authors did not fully accept the review comment and therefore there still remains a lot of things revised for the sake of developing the paper.
Response:
We are truly sorry that our first revised version did not meet your expectations, below please find our response to the original comments
First, an extensive review of previous studies on emotional distress is needed. It is necessary to systematically review the independent variables that may affect emotional distress, and to examine whether these independent variables are reflected in Table 3.
Response: Again, this is a very relevant comment, and we apologize for not addressing it thoroughly earlier. We have added a whole paragraph and a new table 4 in the discussion as examples of studies exploring factors associated with emotional distress in various populations, using diverging designs, samples and assessment scales (line 262, 278). If the editors wish, this table can be displayed in the supplementary instead. In addition to the selection of single studies, we have added the findings form a recent systematic review on impact of covid-19 pandemic on mental health in the general population, and highlight that our recommendation of tailoring interventions based on earlier levels of distress is in line with systematic reviews on both persons with multiple sclerosis and cancer, respectively.
Second, many studies on distress after COVID-19 have been conducted in relation to present study. It is necessary to explain the results by comparing the current research results with the previous research results.
Response: Again, we thank the reviewer for this highly relevant comment. Due to the enormous amount of published papers on the impact of the pandemic on mental health, continuously increasing, we are not able to fully cover and discuss the literature regarding this topic. Nevertheless, we have substantially increased the number of cited and discussed studies, In particular, we have added two studies in the new table 4 on factors associated with emotional distress during the pandemic (ref 31 and 31), and cite one systematic review on the impact of covid-19 on mental health in the general population (32), and discussed our finding in relation to the latter (line 267-274).
Third, the age of the elderly is the main focus of this study. It is necessary to establish a hypothesis and statistically verify it after discussing at the theoretical model regarding distress in the elderly.
Response: We most respectfully argue that this was appropriately addressed in the first revised version, were we added a hypothesis regarding age in the introduction (line 67-70 ),and discussed our finding in light of the hypothesis in the discussion (line 251-253 and 257-258 ).
Fourth, in the conclusion part, it is necessary to suggest what points this study will theoretically contribute and whether future research is necessary.
Response: Thank you, in the first revised version we added a sentence on how our results nuance previous reports on emotional reactivity among elderly (line 360-361) , and that future research should explore the long term impact of the outbreak on mental health in elderly people not readily responding to online surveys, such as those with cognitive impairment and dementia (line 364-367). In this second revised version, we have also added suggestions for future research on strategies for recovery from the outbreak (365), and added three references on mental heath care and the psychiatrists role in this process, as suggested by the academic editor (ref 47-49).